# DUAL-PHASE WHITENING FOR TEST-TIME ADAPTATION

## ABSTRACT

When deploying machine learning models in real-world scenarios, a key challenge is distribution shift where test data differs from the training distribution, often degarding model performance. This problem is particularly challenging in test-time adaptation (TTA), where the model must adapt to unlabeled target data without access to source data or labels. To address this problem, we introduce a novel approach to facilitate target feature learning by utilizing dual-phase whitening (DPW) in connected with whitening Batch Normalization (WBN) and whitening contrastive learning schemes (WCL). WBN operates at the feature transformation level to enforce isotropic feature distributions by ZCA whitening, thereby reducing model dependence on domain-specific covariance structures and improving stability under distribution shifts. WCL extends standard contrastive learning by incorporating global feature whitening, which eliminates redundant feature correlations while enforcing a hyperspherical distribution that better preserves semantic relationships. By the dual-phase whitening, WBN handles low-level feature standardization while WCL optimizes global representation geometry. Thus, we can obtain more generalized features from dual-phase whitening. Our method achieves state-of-the-art performance on major benchmarks including VisDA-C, DomainNet-126, ImageNet-C and CIFAR-100C have several advantages over existing works.

## 1 INTRODUCTION

Deep learning models have achieved remarkable success in computer vision when training and test data share the same distribution, as demonstrated in seminal works Zou et al. (2023); Zhao et al. (2019). However, their performance often deteriorates significantly when deployed in real-world settings where domain shift occurs between source (training) and target (test) distributions He et al. (2016); Long et al. (2015). This degradation comes from various factors including environmental variations, sensor differences, and data corruptions that create distributional discrepancies. Current research has identified this sensitivity to domain shift as a fundamental limitation of deep neural networks in practical applications. To bridge this domain gap, researchers have developed several adaptation approaches Farahani et al. (2021); Papageorgiou & Poggio (2000) to handle these problems.

Test time adaptation (TTA) Sun et al. (2020); Niu et al. (2022) is a method that adapt the model by updating its parameters during testing. However, unlike domain generalization (DG), TTA does not necessitate specific modifications during training and only requires the pre-trained source model and unlabeled target data during the testing phase. This makes TTA more practical and generalizable. TTA enables the model to adapt solely based on online unlabeled data, which can result in a domain gap between the source and target data, rendering the model trained on the source data incompatible with the target data. The majority of previous works on TTA can be viewed from the perspective of the proposed representation revision approach. Some methods adapt the source model by performing a feature alignment process, such as feature matching and prediction adjustment Wang et al. (2021b), or conducting a self-supervised contrastive learning scheme with an online pseudo labeling refinement, as done in AdaContrast Chen et al. (2022). Other methods strive to make target features more uniform in feature space, including entropy minimization Wang et al. (2021b), prototype adjustment Iwasawa & Matsuo (2021), information maximization Liang et al. (2020), and batch normalization statistics alignment Li et al. (2018).

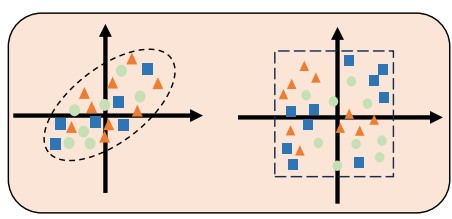 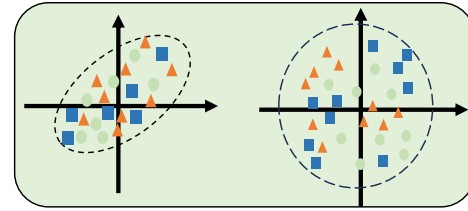

(a) Batch Normalization          (b) Whitening Batch Normalization

Figure 1: Illustrations of multiple normalization methods on centered data. (a) BN performs standardization by stretching/squeezing the data along the axes, such that each dimension has a unit variance. (b) Whitening BN performs ZCA whitening by stretching/squeezing the data along the eigenvectors, such that the covariance matrix is identical.

However, none of these methods address the Test-Time Adaptation (TTA) problem from the perspective of feature whitening. Feature whitening is a powerful technique that transforms the input features into a decorrelated and normalized space, reducing domain-specific biases and enhancing the model's ability to generalize across diverse data distributions. By whitening the features, we aim to enhance the model's generalizability by emphasizing the underlying invariant patterns and structures that are consistent across different domains. This process enables the model to learn more robust and transferable representations that are less sensitive to domain shifts or variations, thereby improving performance in unseen target domains. Motivated by this insight, we propose to improve the model's generalization capability through feature whitening. Specifically, we introduce two key components derived from the properties of feature whitening in TTA including Whitening Batch Normalization (WBN) and Whitening Contrastive Learning (WCL). Unlike standard Batch Normalization (BN), which only centers and scales activations, WBN further decorrelates the features by whitening them in the latent space. We adopt the whitening Batch Normalization approach proposed by Huang et al. Huang et al. (2018), which not only normalizes the target features but also projects them into a common spherical distribution (see Figure 1). This transformation ensures that the features are invariant to linear transformations, thereby reducing domain-specific biases and improving adaptation to new domains. Meanwhile, we observe that whitening contrastive learning can be seamlessly integrated with WBN to further enhance the model's generalization ability. To this end, we incorporate a whitening contrastive loss Ermolov et al. (2021), which encourages the model to learn representations where samples are uniformly distributed on a unit hypersphere. Specifically, this loss function scatters all sample representations into a spherical distribution while penalizing positive pairs that are distant from each other in the embedding space. The whitening operation plays a crucial role in preventing degenerate solutions where all representations collapse into a single point. By combining WBN and WCL, our approach ensures that the model adapts more effectively to unseen data distributions. The decorrelation effect of WBN removes redundant feature dependencies, while WCL promotes a well-structured feature space where samples are discriminatively yet uniformly distributed. Together, these whitening strategies enable the model to generalize better across domains, making it more robust to distribution shifts encountered during test-time adaptation.

Finally, we demonstrate the effectiveness of our proposed approach on commonly used VisDA-C Peng et al. (2017), DomainNet-126 Peng et al. (2019), ImageNet-C Hendrycks & Dietterich (2019) and CIFAR 100-C Hendrycks & Dietterich (2019). We summarize our contributions as follows:

- The use of whitening Batch Normalization (WBN) is an improvement over the original Batch Normalization (BN) technique, as it enhances the training speed and generalization performance.

- We propose utilizing the whitening contrastive learning (WCL) loss function. WCL ensures that the batch samples lie in a spherical distribution and serves as an alternative to positive-negative instance contrasting methods.

- Our experiments show that the proposed method outperforms existing test time adaptation approaches on domain generalization benchmarks.

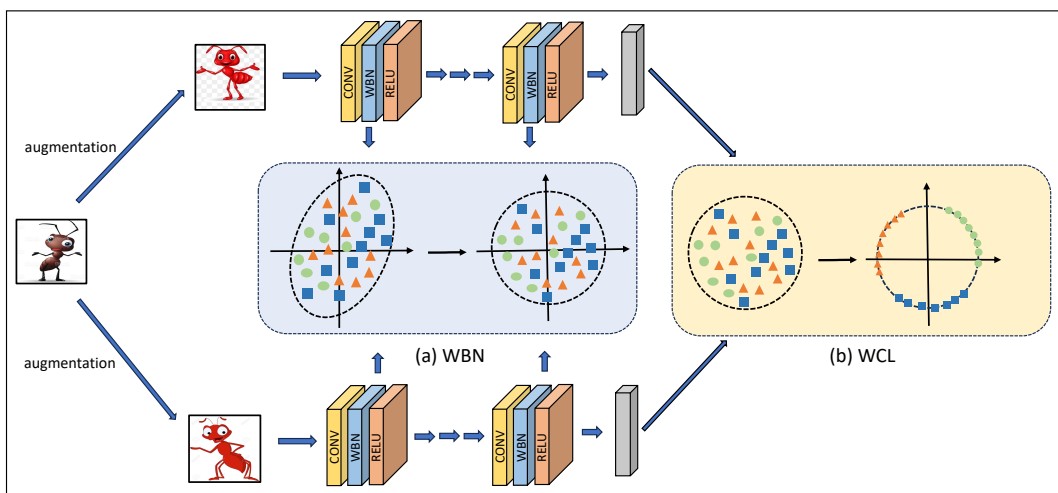

Figure 2: **Framework of our Dual-Phase Whitening for test-time adaptation (DPW):** At the beginning of adaptation, the model and momentum model are initialized by the source model. A target image is transformed by one weak and two strong augmentations. We leverage AdaContrast as our baseline and revise the two modules, including (a) Whitening Batch Normalization (WBN) and (b) Whitening Contrastive Learning (WCL), respectively.

## 2 METHOD

### 2.1 MOTIVATION AND PRELIMINARIES

This work presents a systematic investigation into feature whitening for Test-Time Adaptation (TTA) Liang et al. (2025); Karmanov et al. (2024); Farina et al. (2024), a critical yet underexplored direction in domain adaptation research. While conventional TTA methods primarily focus on statistical alignment or entropy minimization, we identify feature whitening as a powerful but neglected mechanism to address the fundamental challenge of distribution shift. Our proposed Dual-Phase Whitening (DPW) framework introduces two complementary whitening strategies that operate at different levels of feature representation to maximize model generalizability. The first phase, Whitening Batch Normalization (WBN), performs ZCA whitening at the feature transformation level to enforce isotropic feature distributions (Figure 2(a)). Unlike standard batch normalization that merely standardizes activations, WBN additionally decorrelates features by: (1) computing the covariance matrix of batch activations, (2) performing eigen-decomposition to obtain principal components, and (3) projecting features onto a unit sphere through whitening transformation. This process effectively reduces the model's dependence on domain-specific covariance structures while improving numerical stability during adaptation. The second phase, Whitening Contrastive Learning (WCL), operates in the latent representation space to enforce a hyperspherical distribution of embeddings (Figure 2(b)). Building upon contrastive learning principles, WCL introduces two key innovations: (a) a global whitening constraint that eliminates redundant feature correlations through batch-wise ZCA whitening, and (b) a distance metric learning objective that preserves semantic relationships while preventing representation collapse. The combined effect ensures that similar samples cluster together while maintaining uniform angular distribution across all classes. Our DPW approach draws inspiration from established whitening techniques like DBN Huang et al. (2018) and WSS Ermolov et al. (2021), but makes crucial advancements for the TTA setting. While previous works demonstrated whitening benefits for linear models (DBN) and representation learning (WSS), they fail to address three critical aspects of real-world TTA: (1) the absence of source data during adaptation, (2) the need for single-pass online processing of test batches, and (3) the risk of instability with small batch sizes. Our framework specifically addresses these challenges through momentum-encoded statistics for WBN and stable covariance estimation for WCL, making it uniquely suited for practical deployment scenarios.

We are tackling the problem of closed-set test-time adaptation in image recognition, where the adaptation process does not utilize any source data. The source model is trained on pairs of images and labels $\{x_s^i, y_s^i\}_{i=1}^{n_s} \in \mathcal{D}_s$, where $x_s^i \in \mathcal{X}_s$ and $y_s^i \in \mathcal{Y}_s$. Our objective is to adapt the trained source model to unlabeled target data $\{x_t^i\}_{i=1}^{n_t} \in \mathcal{X}_t$, while the corresponding labels $\{y_t^i\}_{i=1}^{n_t} \in \mathcal{Y}_t$ are only accessible for evaluation. In the closed-set scenario, both the source and target domains have the same label space $\mathcal{Y}_s = \mathcal{Y}_t = \mathcal{Y}$. This assumption differentiates our work from more challenging open-set or partial-set adaptation scenarios, where the target domain may contain novel categories not present in the source domain. The preservation of identical label spaces enables reliable knowledge transfer while maintaining well-defined evaluation protocols.

## 2.2 Whitening Batch Normalization (WBN)

### 2.2.1 Batch Normalization

For a Batch Normalization (BN) layer that takes as its input a batch of $D$-dimensional vectors $X = (x_1, \cdots, x_B) \in \mathcal{R}^{D \times B}$, its output is a batch of vectors $Y = (y_1 \cdots, y_B) \in \mathcal{R}^{D \times B}$, computed as follows

$$y_{i,k} = \frac{x_{i,k} - \mu_k}{\sqrt{\sigma_k^2 + \epsilon}} \cdot \gamma_k + \beta_k$$

for all $i \in \{1, \cdots, B\}$ and $k \in \{1, \cdots, D\}$, where $\gamma, \beta$ are learnable affine parameters, $\epsilon$ is a small constant originally proposed for numerical stability. In training time, $\mu_k, \sigma_k^2$ are mean and variance computed over the $k$-th row of the input batch $X$, and in inference time, running estimations from training time is used.

Dimensional collapse can harm utility and should be addressed appropriately. By definition, dimensional collapse is associated with strong correlations between axes. As a sanity check, we adapt DBN Huang et al. (2018) to standardize the covariance matrix for mitigation of this issue.

### 2.2.2 Domain-specific Whitening Transform

As previously mentioned, BN involves standardizing each sample $\mathbf{x}_i \in B$ based on its dimensions. However, this means that even after normalization, the features of batch samples may still be correlated. We aim to address the domain-shift problem with feature normalization, and therefore argue that simple standardization is insufficient for aligning the source and target marginal distributions. To better achieve this alignment, we suggest utilizing Batch Whitening (BW) as an alternative to BN. Essentially, BW involves transforming the data to have zero means and unit variances, while also decorrelating the features via principal component analysis.

$$BW(x_{i,k}; \Omega) = \gamma_k \hat{x}_{i,k} + \beta_k, \tag{1}$$

$$\hat{\mathbf{x}}_i = W_B(\mathbf{x}_i - \boldsymbol{\mu}_B). \tag{2}$$

In Eq. equation 2, $\boldsymbol{\mu}_B$ is the mean of the elements in batch $B$, and the matrix $W_B$ is determined such that $W_B^\top W_B = \Sigma_B^{-1}$, where $\Sigma_B$ is the covariance matrix computed using $B$. $\Omega = (\boldsymbol{\mu}_B, \Sigma_B)$ represents the first and second-order statistics specific to the batch. Eq. equation 2 is utilized to whiten $\mathbf{x}_i$, resulting in a set of vectors $\hat{B} = \{\hat{\mathbf{x}}_1, ..., \hat{\mathbf{x}}_m\}$ that are distributed spherically.

Our network takes as input the batches of target data, randomly extracted from $\mathcal{X}_t$. Specifically, given any arbitrary layer $l$ in the network, let $B^t = \{\mathbf{x}_1^t, ..., \mathbf{x}_k^t\}$ denote the batch of intermediate output activations from layer $l$. Using Eq. equation 1-equation 2 we can now define our WBN. Let $x^s$ denote the inputs to the WBN layer from the target domain. Our WBN is defined as follows

$$WBN(x^t; \Omega^t) = BW(x^t, \Omega^t). \tag{3}$$

$B^t$ and use them for whitening the corresponding activations, projecting the batches into a common spherical distribution (Figure. 2 (a)).

To compute $W_B^t$, we adopt the method proposed in Roy et al. (2019), which employs Cholesky decomposition Higham (1990). Unlike the approach taken in Huang et al. (2018), we eliminate the coloring step following the whitening process and instead apply simple scale and shift operations

to avoid introducing additional network parameters. Additionally, we utilize feature grouping to enhance the accuracy of batch-statistics estimation when the number of samples $m$ is small and the feature dimension $d$ is large, distinguishing our method from Huang et al. (2018). During training, the WBN layers continually update the target domain statistics through a moving average of the batch statistics.

## 2.3 Whitening Contrastive Learning (WCL)

Our approach involves extracting an embedding $\mathbf{z} = f(x; \theta)$ from an input image $x$ using an encoder network $f(\cdot; \theta)$ with adjustable parameters $\theta$ (further explained below). To ensure optimal performance, we require that (1) the image embeddings are not drawn from a degenerate distribution that could cause all representations to converge to a single point, and (2) positive image pairs $(x_i, x_j)$ sharing similar semantics should cluster closely together. We have formulated this problem as follows

$$min_\theta \, \mathbb{E}[dist(\mathbf{z}_i, \mathbf{z}_j)], \tag{4}$$
$$s.t. \, cov(\mathbf{z}_i, \mathbf{z}_i) = cov(\mathbf{z}_j, \mathbf{z}_j) = I, \tag{5}$$

We utilize Eq. 5 to constrain the distribution of the $\mathbf{z}$ values in order to prevent a degenerate distribution where all probability mass concentrates into a single point. Here, $dist(\cdot)$ represents the distance between vectors, $I$ is the identity matrix, and $(\mathbf{z}_i, \mathbf{z}_j)$ corresponds to a positive pair of images $(x_i, x_j)$. Additionally, Eq. 5 imposes linear independence among all components of $\mathbf{z}$, encouraging each dimension of $\mathbf{z}$ to represent different semantic content. We adopt the cosine similarity for distance calculation and implement it through mean squared error between normalized vectors.

$$
\begin{aligned}
dist(\mathbf{z}_i, \mathbf{z}_j) &= \left\| \frac{\mathbf{z}_i}{\|\mathbf{z}_i\|_2} - \frac{\mathbf{z}_j}{\|\mathbf{z}_j\|_2} \right\|_2^2 \\
&= 2 - 2 \frac{\langle \mathbf{z}_i, \mathbf{z}_j \rangle}{\|\mathbf{z}_i\|_2 \cdot \|\mathbf{z}_j\|_2}
\end{aligned}
\tag{6}
$$

To obtain positive samples with shared semantics, we follow a procedure similar to Ermolov et al. (2021), utilizing standard image augmentation techniques within AdaContrast. Specifically, we randomly and independently select parameters $\mathbf{p}$ for each positive sample extracted from a single image $x$ and $x_i = T(x; \mathbf{p}_i)$. We use the notation $pos(i, j)$ to indicate that $x_i$ and $x_j$ have been extracted from the same image.

The number of positive samples per image denoted as $d$, may vary, with the trade-off between diversity in the batch and the amount of training signal. Many methods opt for only one positive pair ($d = 2$), prioritizing more negative pairs. However, W-MSE Ermolov et al. (2021) has demonstrated improved performance using five samples. In our MSE-based loss approach, we utilize all possible $d(d-1)/2$ combinations of positive samples. We conduct experiments using $d = 2$ (one positive pair) and $d = 4$ (6 positive pairs).

To facilitate representation learning, we employ an unsupervised backbone encoder network $E(\cdot)$ within AdaContrast. We use a conventional ResNet He et al. (2016) as the encoder, with $\mathbf{h} = E(x)$ representing the output of the average-pooling layer. This approach is straightforward and easily reproducible, compared to other methods that use encoder architectures specifically tailored to a particular pretext task. As $\mathbf{h} \in \mathbb{R}^{512}$ or $\mathbf{h} \in \mathbb{R}^{2048}$ is a high-dimensional vector, we follow Ermolov et al. (2021) and employ a nonlinear projection head $g(\cdot)$ to project $\mathbf{h}$ into a lower-dimensional space: $\mathbf{v} = g(\mathbf{h})$. $g(\cdot)$ is implemented using a multi-layer perceptron (MLP) with a hidden layer and a BN layer, and the complete network $f(\cdot)$ is the composition of $g(\cdot)$ with $E(\cdot)$.

Let $N$ represent the number of original images, and consider a batch of samples $B = x_1, ..., x_k$, where $K = Nd$. We obtain the corresponding batch of features, denoted by $V = \{\mathbf{v}_1, ...\mathbf{v}_k\}$, using the aforementioned approach. In our proposed W-MSE loss, we calculate the mean squared error (MSE) across all $Nd(d-1)/2$ positive pairs, satisfying the constraint through reparameterization of the $\mathbf{v}$ variables with the whitened variables $\mathbf{z}$

$$L_{WCL}(V) = \frac{2}{Nd(d-1)} \sum dist(\mathbf{z}_i, \mathbf{z}_j), \tag{7}$$

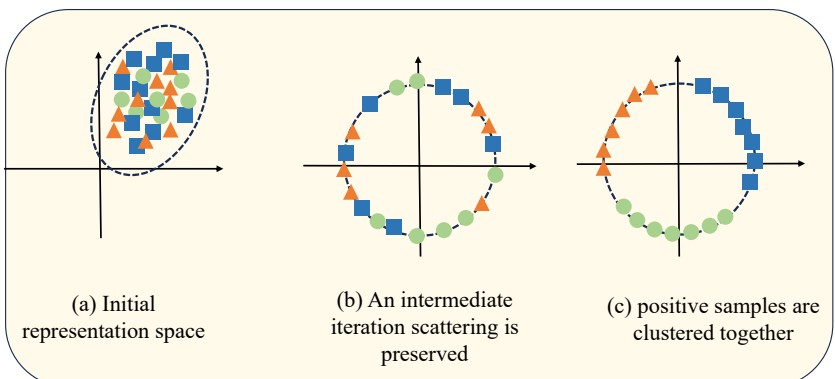

(a) Initial representation space

(b) An intermediate iteration scattering is preserved

(c) positive samples are clustered together

Figure 3: A schematic representation of the WCL-based optimization process. Positive pairs are indicated with the same shapes and colors. (a) A representation of the batch features in $V$ when training starts. (b) The WCL computed over the normalized $\mathbf{z}$ features encourages the network to move the positive pair representations closer to each other. (c) The subsequent iterations move closer and closer to the positive pairs, while the relative layout of the other samples is forced to lie in a spherical distribution.

Here, the sum is over all $(\mathbf{v}_i, \mathbf{v}_j) \in V$, with $pos(i, j) = true$, and $\mathbf{z} = Whitening(\mathbf{v})$ representing the whitened variables. Specifically, we have

$$Whitening(\mathbf{v}) = W_V(\mathbf{v} - \boldsymbol{\mu}_V). \tag{8}$$

Eq. 8 defines $\boldsymbol{\mu}_V$ as the mean of the elements in $V$: $\boldsymbol{\mu}_V = \frac{1}{K} \sum_k \mathbf{v}_k$. The matrix $W_V$ is determined such that $W_V^\top W_V = \Sigma_V^{-1}$, where $\Sigma_V$ denotes the covariance matrix of $V$.

$$\Sigma_V = \frac{1}{K - 1} \sum_k (\mathbf{v}_k - \boldsymbol{\mu}_V)(\mathbf{v}_k - \boldsymbol{\mu}_V)^T. \tag{9}$$

The proposed loss function in Eq. 7 operates on the premise that it penalizes distant positive pairs, thereby encouraging $g(E(\cdot))$ to reduce their inter-positive distances. Meanwhile, since $Z$ must conform to a spherical distribution, the remaining samples must be repositioned to satisfy constraint equation 5, as illustrated in Figure 3.

## 2.4 Overall Formulation

With the whitening features, we can define our final objective for effective test-time adaptation. First, a recent work of Contrast Test-Time Adaptation (AdaContrast) Chen et al. (2022), which proposes a novel way to leverage self-supervised contrastive learning to facilitate target feature learning, along with an online pseudo labeling scheme with a refinement that significantly denoises pseudo labels. AdaConstrast consists of the exclusion of same-class negative pairs loss $\mathcal{L}^{\text{ctr}}$, weak-strong consistency loss $\mathcal{L}^{\text{ce}}$ and diversity regularization loss $\mathcal{L}^{\text{div}}$. Combining all these loss functions together, we can get the objective of AdaContrast as

$$\mathcal{L}_{\text{Adacontrast}} = \mathcal{L}^{\text{ctr}} + \mathcal{L}^{\text{ce}} + \mathcal{L}^{\text{div}} \tag{10}$$

Then the WBN and WCL module is applied for AdaContrast, our final objective becomes

$$\mathcal{L}_{\text{DPW}} = \mathcal{L}_{\text{Adacontrast}}^{\text{Whitening}} \tag{11}$$

## 3 Experiment

We conduct experiments of closed-set adaptation on major benchmarks. In the following, we first compare the proposed method DPW with the previous state-of-the-art algorithms. Then, we discuss several desirable test-time properties of DPW, followed by ablation and analysis of the important design modules that brought the gains.

Table 1: Classification accuracy (%) on VisDA-C train → val. All methods use ResNet-101 backbone except the on-target rows, which use ResNet-18 as the student network. Bold is the highest. The proposed AdaContrast surpasses the previous state-of-the-art by 1.0% Avg. When applying an extra knowledge distillation stage following Wang et al. (2021a), we achieve the highest 87.8% with a small ResNet-18 backbone.

| Method | source-free | plane | bcycl | bus | car | horse | knife | mcycl | person | plant | sktbrd | train | truck | Avg. |
|---|---|---|---|---|---|---|---|---|---|---|---|---|---|---|
| DANN | no | 81.9 | 77.7 | 82.8 | 44.3 | 81.2 | 29.5 | 65.1 | 28.6 | 51.9 | 54.6 | 82.8 | 7.8 | 57.4 |
| CDAN | no | 85.2 | 66.9 | 83.0 | 50.8 | 84.2 | 74.9 | 88.1 | 74.5 | 83.4 | 76.0 | 81.9 | 38.0 | 73.9 |
| CDAN+BSP | no | 92.4 | 61.0 | 81.0 | 57.5 | 89.0 | 80.6 | 90.1 | 77.0 | 84.2 | 77.9 | 82.1 | 38.4 | 75.9 |
| CAN | no | 97.0 | 87.2 | 82.5 | 74.3 | 97.8 | 96.2 | 90.8 | 80.7 | 96.6 | 96.3 | 87.5 | 59.9 | 87.2 |
| SWD | no | 90.8 | 82.5 | 81.7 | 70.5 | 91.7 | 69.5 | 86.3 | 77.5 | 87.4 | 63.6 | 85.6 | 29.2 | 76.4 |
| MCC | no | 88.7 | 80.3 | 80.5 | 71.5 | 90.1 | 93.2 | 85.0 | 71.6 | 89.4 | 73.8 | 85.0 | 36.9 | 78.8 |
| Source only | - | 57.2 | 11.1 | 42.4 | 66.9 | 55.0 | 4.4 | 81.1 | 27.3 | 57.9 | 29.4 | 86.7 | 5.8 | 43.8 |
| MA | yes | 94.8 | 73.4 | 68.8 | 74.8 | 93.1 | 95.4 | 88.6 | 84.7 | 89.1 | 84.7 | 83.5 | 48.1 | 81.6 |
| BAIT | yes | 93.7 | 83.2 | 84.5 | 65.0 | 92.9 | 95.4 | 88.1 | 80.8 | 90.0 | 89.0 | 84.0 | 45.3 | 82.7 |
| SHOT | yes | 95.3 | 87.5 | 78.7 | 55.6 | 94.1 | 94.2 | 81.4 | 80.0 | 91.8 | 90.7 | 86.5 | 59.8 | 83.0 |
| + On-target | yes | 96.0 | 89.5 | 84.3 | 67.2 | 95.9 | 94.2 | 91.0 | 81.5 | 93.8 | 89.9 | 89.1 | 58.2 | 85.9 |
| AdaContrast (baselines) | yes | 97.0 | 84.7 | 84.0 | 77.3 | 96.7 | 93.8 | 91.9 | 84.8 | 94.3 | 93.1 | 94.1 | 49.7 | 86.8 |
| DPW | yes | 97.6 | 87.8 | 87.3 | 82.5 | 96.9 | 92.8 | 94.1 | 87.2 | 95.3 | 91.2 | 92.6 | 48.5 | 87.8 |

Table 2: Classification accuracy (%) on 7 domain shifts of DomainNet-126. All methods use ResNet-50 backbone. Bold is the highest. The proposed DPW achieves the highest average performance.

| Method | Source-free | R→C | R→P | P→C | C→S | S→P | R→S | P→R | Avg. |
|---|---|---|---|---|---|---|---|---|---|
| MCC | no | 44.8 | 65.7 | 41.9 | 34.9 | 47.3 | 35.3 | 72.4 | 48.9 |
| Source only | - | 55.5 | 62.7 | 53.0 | 46.9 | 50.1 | 46.3 | 75.0 | 55.6 |
| TENT | yes | 58.5 | 65.7 | 57.9 | 48.5 | 52.4 | 54.0 | 67.0 | 57.7 |
| SHOT | yes | 67.7 | 68.4 | 66.9 | 60.1 | 66.1 | 59.9 | 80.8 | 67.1 |
| AdaContrast (baseline) | yes | 70.2 | 69.8 | 68.6 | 58.0 | 65.9 | 61.5 | 80.5 | 67.8 |
| DPW | yes | 72.3 | 70.4 | 69.7 | 61.5 | 66.2 | 63.7 | 81.2 | 69.2 |

Table 3: Ablation study on DomainNet-126. AdaContrast denotes the original baseline. AdaContrast (+WBN) denotes whitening Batch Normalization applied on the baseline AdaContrast. AdaContrast (+WCL) denotes whitening contrastive learning applied on the baseline AdaContrast.

| Method | R→C | R→P | P→C | C→S | S→P | R→S | P→R | Avg. |
|---|---|---|---|---|---|---|---|---|
| AdaContrast | 70.2 | 69.8 | 68.6 | 58.0 | 65.9 | 61.5 | 80.5 | 67.8 |
| AdaContrast (+WBN) | 71.0 | 70.1 | 68.9 | 59.3 | 66.0 | 62.7 | 80.8 | 68.4 |
| AdaContrast (+WCL) | 71.4 | 70.2 | 69.3 | 60.9 | 66.1 | 62.8 | 80.7 | 68.7 |
| DPW | 72.3 | 70.4 | 69.7 | 61.5 | 66.2 | 63.7 | 81.2 | 69.2 |

## 3.1 EXPERIMENTAL SETUP

**Datasets and Metrics.** We use VisDA-C Peng et al. (2017), DomainNet-126 Peng et al. (2019), ImageNet-C Hendrycks & Dietterich (2019) and CIFAR 100-C Hendrycks & Dietterich (2019) for evaluating our method and comparison.

VisDA-C Peng et al. (2017) is a large-scale and challenging benchmark, primarily designed for the 12-class synthesis-to-real object recognition task. The dataset consists of a source domain with 152,000 synthetic images generated by rendering 3D models and a target domain with 55,000 real object images sampled from Microsoft COCO.

DomainNet-126 is a subset of DomainNet Peng et al. (2019), which is currently the largest unsupervised domain adaptation dataset. It consists of six distinct domains and approximately 0.6 million images distributed among 345 categories. For our evaluation, we follow the approach of Saito et al. (2019) and select 126 classes from four domains, namely Real, Clipart, painting, and sketch.

ImageNet-C and CIFAR-100C Hendrycks & Dietterich (2019) were initially designed to evaluate the robustness of classification networks. Each dataset consists of 15 different corruption types, each with five severity levels. These corruptions were applied to images from the test set of the clean ImageNet-C or CIFAR-100 datasets.

It is important to note that the original DomainNet dataset has noisy labels. Therefore, we employ a subset of the dataset containing 126 classes from four domains, namely Real, Sketch, Clipart, and Painting, as recommended by the authors' follow-up work Saito et al. (2019). This subset is

Table 4: Classification accuracy (%) on 12 domain shifts of ImageNet-C and CIFAR-100C.

| Method (ImageNet-C) | Gauss | Shot | Impul | Defoc | Glass | Motion | Zoom | Snow | Frost | Fog | Brit | Cont |
|---|---|---|---|---|---|---|---|---|---|---|---|---|
| Tent | 53.9 | 53.9 | 55.3 | 55.9 | 51.9 | 59.8 | 52.6 | 58.7 | 61.4 | 71.3 | 78.2 | 68.0 |
| Tent + Whitening | **54.5** | **54.4** | **56.8** | **56.1** | **52.6** | **60.7** | **54.0** | **59.6** | **63.4** | **72.5** | **79.4** | **69.5** |
| CoTTA | 43.5 | 44.1 | 45.4 | 46.8 | 51.5 | 47.6 | 36.8 | 48.3 | 39.7 | 43.5 | 47.7 | 42.1 |
| CoTTA + Whitening | **44.9** | **42.5** | **46.7** | **37.9** | **39.3** | **48.3** | **43.7** | **45.7** | **43.5** | **63.6** | **75.2** | **63.7** |
| MEMO | 40.6 | 37.5 | 40.6 | 33.4 | 26.7 | 41.2 | 35.4 | 54.7 | 67.1 | 65.9 | 66.7 | 59.3 |
| MEMO + Whitening | **41.5** | **38.6** | **41.3** | **34.7** | **28.3** | **36.5** | **29.4** | **34.5** | **55.6** | **68.3** | **56.4** | **60.4** |
| Method (CIFAR-100C) | Gauss | Shot | Impul | Defoc | Glass | Motion | Zoom | Snow | Frost | Fog | Brit | Cont |
| RoTTA | 31.8 | 36.7 | 40.9 | 42.1 | 30.3 | 27.9 | 25.4 | 32.3 | 34.0 | 38.8 | 38.3 | 31.3 |
| RoTTA + Whitening | **32.5** | **37.3** | **41.5** | **43.7** | **33.5** | **34.8** | **28.9** | **26.4** | **33.2** | **35.3** | **39.7** | **39.5** |

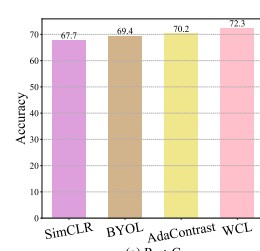 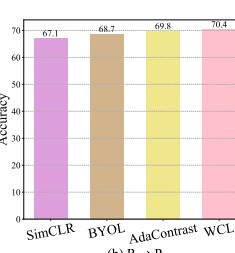 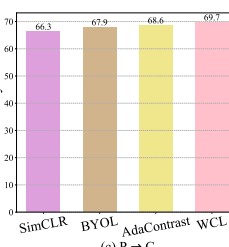 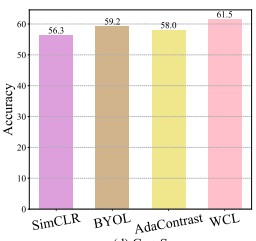

Figure 4: Classification accuracy on DomainNet-126 with different contrastive loss. (a) R → C, (b) R → P, (c) P → C, (d) C → S.

referred to as DomainNet-126. We evaluate the performance of the methods on seven domain shifts, as constructed from the domains, following the approach of Saito et al. (2019). We report the top-1 accuracy under each domain shift as well as the average across all seven shifts. For VisDA-C, we compare per-class top-1 accuracies and their average.

## 3.2 RESULTS

**VisDA-C.** Table. 1 compares AdaContrast with state-of-the-art unsupervised domain adaptation and test-time adaptation methods on VisDA-C's "train" to "val" shift. For UDA, our method is on-par with a strong UDA baseline CAN Kang et al. (2019) and significantly outperforms the others by a large margin, even though we do not utilize source data at all during test-time adaptation. In the more challenging TTA setting, we achieve the highest per-class average accuracy by a notable margin (+1.0%) upon AdaContrast. This performance gap is especially significant given that VisDA-C represents one of the most challenging large-scale domain adaptation benchmarks, with substantial domain shift across all classes.

**DomainNet-126.** Table. 2 shows the comparison between DPW and state-of-the-art UDA (first section) and TTA (second sections) methods. Without needing source data during the adaptation, our method outperforms the UDA method MCC Jin et al. (2020) by +20.3% on the averaged performance. This significant margin highlights the effectiveness of our approach in handling domain shifts without relying on source data, unlike traditional UDA techniques. When being compared to TTA methods DPW outperforms TENT by +11.5% on the averaged performance. It achieves the best performance on four out of seven domain shifts as well as the highest averaged performance.

**ImageNet-C and CIFAR 100-C.** Table 4 demonstrates that integrating DPW with existing TTA methods consistently enhances performance across a wide range of corruptions and datasets, particularly on ImageNet-C and CIFAR-100-C. For ImageNet-C, the Tent baseline shows moderate performance with an average accuracy of 61.2%. However, when whitening is added, the accuracy increases to 62.3%, with the largest gains observed in Motion, Fog, Brightness, and Contrast corruptions. This indicates that whitening enhances the model's robustness to distortions, improving its adaptability. Similarly, CoTTA, with its baseline performance of 49.1%, benefits from whitening, achieving an average accuracy of 50.3%. The improvement is particularly evident in Fog, Brightness, and Motion corruptions. The MEMO baseline, which initially performs at 43.2%, also sees a notable increase

when combined with whitening, reaching an average accuracy of 44.2%, with improvements in Fog, Brightness, and Motion.

When evaluated on CIFAR-100-C, RoTTA shows relatively low performance with an average accuracy of 35.0%. However, adding whitening boosts the accuracy to 36.1%, with significant gains in Motion, Brightness, and JPEG corruptions. These results confirm that whitening is effective in mitigating the adverse effects of domain shifts and corruptions in a variety of challenging scenarios. Overall, the integration of whitening consistently improves the robustness and adaptability of existing TTA methods, providing significant performance boosts across diverse corruption types and domain shifts in both ImageNet-C and CIFAR-100-C datasets. These improvements highlight the generalizability of the method and its potential for real-world applications where source data is unavailable during adaptation.

### 3.3 ANALYSIS

**Ablation Study.** We performed an ablation study on the DomainNet-126 dataset to assess the effectiveness of each component in our DPW. Table. 3 presents the classification results of the two modules of our method. We established a baseline result with WBN and WCL modules. We removed WBN and WCL individually to understand how the whitening module impacts the learning framework. Our observations indicate that the accuracy of the model with WBN (WCL) module is 0.6% (0.9%) higher than the baseline. Additionally, compared to the baseline, there is a clear performance gain of 1.4%. This implies that whitening from the same class closer in the embedding space is beneficial for improving the model's generalization power.

**Domain Generalization Tasks.** To evaluate the effectiveness of the proposed method on domain generalization, we conduct experiments over Digits-DG tasks. We use a recently proposed method SADML Wang et al. (2022a) as our baseline and apply the proposed method WBN and WCL to SADML. Specifically, on the hardest target domains SVHN and SYN, which involve cluttered digits and low image qualities, our method outperforms SADML 2.13% and 1.77%, respectively. The success of our method indicates that the whitening module applied to the domain generalization task can significantly promote its performance on leave-one-domain-out images.

**Compared with Different Contrastive Loss.** We show the DomainNet-126 results with different contrastive loss, and we compare WCL with results of other state-of-the-art approaches of contrastive learning such as SimCLR Chen et al. (2020a), BYOL Grill et al. (2020), AdaContrast, Despite some configuration details are different, in all cases, the encoder is a ResNet. As illustrated in Figure 4, WCL achieves highly competitive performance despite two key advantages: (1) we do not perform intensive hyperparameter tuning, and (2) our framework is significantly simpler than many existing approaches (e.g., it avoids the asymmetric architectures or momentum encoders used in BYOL). These results suggest that WCL effectively balances performance and simplicity, making it a practical choice for contrastive representation learning.

## 4 CONCLUSION

We introduced DPW, a novel test-time adaptation approach for closed-set DA in image classification. In this paper, we tackle the challenge of adapting to a new domain without labeled data or access to the source domain. To address this problem, we propose Test-time Adaptation (TTA) as a solution that enables our model to adapt to the new data distribution during testing, using only unlabeled test data batches. Our approach introduces a novel method to facilitate target feature learning, which involves utilizing dual-phase whitening (DPW) in conjunction with whitening Batch Normalization and whitening contrastive learning schemes. To enhance the optimization efficiency and generalization capabilities of our models, we employ ZCA whitening for feature decorrelation. Additionally, we enforce the constraint that batch samples should conform to a spherical distribution by whitening the latent space features in whitening contrastive learning. Our method achieves state-of-the-art performance on prominent benchmarks such as VisDA-C and DomainNet-126, and offers several advantages over existing works.

## 5 ETHICS STATEMENT

This work complies with the ICLR Code of Ethics. We present DPW, a framework for test-time adaptation, evaluated on publicly available benchmark datasets. These datasets contain no personally identifiable or sensitive information, ensuring no risks to privacy or security. Our research advances energy-efficient test-time adaptation with potential benefits for scientific and technological applications. All experimental protocols are transparently documented, with fair comparisons to prior work. The contributions are intended solely for research, supporting AI development.

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

# APPENDICES

## A  RELATED WORK

**Test-time Adaptation** In this setting, the unlabeled target data is not available during the training phase of the source model. Instead, the model only has access to the target data when deployed in the target environment, as the stream of test data batches. As a consequence, the unlabeled target data is not available in large quantities, but only in small batches. So far, there are two main lines of work dealing with this online adaptation problem. The first approach is based on the batch normalization layer in the network, with the idea of using the test-time batch statistics for these layers instead of the source statistics. This can partially help with the representation misalignment between the source and target distributions but is only applicable when the network has BatchNorm layers and the batch size is large. The other line of work is to use an unsupervised surrogate loss to "refine" the model on the target data. For example, TENT minimizes the prediction entropy of the model on the target data. However, as pointed out by its authors, TENT needs a large batch size to avoid collapsing to a trivial prediction, due to the characteristics of this surrogate objective. In TENT, this is achieved with a simple entropy minimization loss, which informs the optimization of scale and bias parameters of batch normalization layers. As for batch normalization layers' statistics, they are re-estimated on the test data, similarly to what is done in adaptive batchnorm (AdaBN) methods Hendrycks & Dietterich (2019); Burns & Steinhardt (2021), which have shown strong performance on the perturbations of ImageNet-C Hendrycks & Dietterich (2019). In a similar spirit, Liang *et al.* Liang et al. (2020) updates the parameters of the feature extractor of a given model by maximizing a mutual information objective. Meanwhile, TTT and TTT+++ train a self-supervised task alongside the main task on the source domain and continue to train the self-supervised task during the online adaptation phase. Therefore, TTT and TTT+++ are not fully off-the-shelf test-time adaptation methods, since they require modifications to the training phase of the source domain. Recently, TIPI Nguyen et al. (2023) propose to use an invariance regularize as the surrogate loss during test-time adaptation, motivated by our theoretical results regarding the model's performance under input transformations. There also have been several works Niu et al. (2022) that study datapoints selection strategies for optimizing the unsupervised surrogate loss which only optimize the loss for reliable and informative datapoints.

**Normalization Techniques** It is widely in CNNs to make them learn faster, more stable, and increase their generalization ability. Representative methods include BN Ioffe & Szegedy (2015), Layer Normalization Ba et al. (2016), Adaptive BN Li et al. (2018), Group Normalization Wu & He (2018), Switchable Normalization Shao et al. (2019) and Representation Normalization Lyu & Simoncelli (2008). For better adaptation, researchers have devised novel designs to mitigate the shortcomings in BN. AutoDIAL uses the statistics of two domains channel by channel in order to align the source and target feature distributions. Domain Specific BN Chang et al. (2019) normalizes the source and target representations completely individually, including affine parameters. Transferable Normalization Wang et al. (2019) utilizes the statistics of two domains to calculate corresponding channel attention, which is all detached from the computation graph. ConvNorm Liu et al. (2021) proposes an adaptation layer $\mathcal{A}$ to whiten and color source domain data, then $\mathcal{A}$ is fine-tuned on the target domain. DWT Roy et al. (2019) uses two co-variance matrices to whiten feature maps from source and target domains, respectively. TTN Lim et al. (2023) employs test batch statistics mitigates the performance degradation caused by the domain shift and present a strategy for mixing conventional batch normalization and transductive batch normalization based on the interpolating weight derived from the optimization procedure utilizing the sensitivity to domain shift and show that our method significantly outperforms other normalization techniques in various realistic evaluation settings. DomainAdapter Zhang et al. (2023) dynamically fuses statistics between source and test batch statistics for an accurate statistics estimation and generalizatized entropy minimization loss effectively enhances the adaptation ability. Different from these existing normalization counterparts, we focus on decoupling batch normalization in CNNs for TTA. In this paper, we leverage Decoupling Batch Normalization (DBN), which extends Batch Normalization to include whitening over mini-batch data.

## B    COMPARED BASELINE

We perform a comparative analysis of our method against both classical unsupervised domain adaptation (UDA) baselines and source-free/test-time adaptation (TTA) baselines. For UDA methods, we compare our approach to DANN Ganin & Lempitsky (2015), CDAN Long et al. (2017), CDAN+BSP Chen et al. (2019), CAN Kang et al. (2019), SWD Lee et al. (2019), and MCC Jin et al. (2020). It is important to note that all UDA methods have access to source data during adaptation. For TTA methods, we compare our approach to MA Li et al. (2020), BAIT Yang et al. (2020), TENT Wang et al. (2021b), SHOT Liang et al. (2020), and On-target Wang et al. (2021a), CoTTA Wang et al. (2022b), MEMO Zhang et al. (2022) and RoTTA Yuan et al. (2023). These methods represent a variety of approaches, including image generation, class prototypes, entropy minimization, pseudo labeling, and the combination of contrastive features and pseudo labeling.

## C    MODEL ARCHITECTURE

Our proposed method assumes a general architecture with a feature encoder followed by a classifier. For the purpose of comparison, we employ ResNet models He et al. (2016) as our backbones in various experiments. We use AdaContrast Chen et al. (2022) as our baseline and introduce a 256-dimensional bottleneck consisting of a fully-connected layer. To modify ResNet, we replace the first Batch Normalization (BN) layer and the BN layers in the first residual block with Weight Standardization (WBN) layers. We adopt the implementation and hyperparameter configuration from the AdaContrast method. As we utilize a lower-dimensional bottleneck, we remove the original projection heads from MoCo Chen et al. (2020b) without any observed performance decline.

## D    IMPLEMENTATION

We adopt Pytorch Paszke et al. (2019) for all our implementations. For source training, we initialize the ResNet backbone with ImageNet-1K Deng et al. (2009) pre-trained weights from the Pytorch model zoo. We follow the approach of Liang et al. (2020) and randomly split the source data into a 9:1 ratio, with 90% used for training the source model and 10% used for validation. Source training is conducted for 10 and 60 epochs for VisDA-C and DomainNet-126, respectively. For target training, we use only 15 epochs for all datasets, unless otherwise indicated. For all experiments, we utilize the SGD optimizer with momentum 0.9 and weight decay 1e-4, and cosine annealing on the learning rate. The learning rate is decayed from its initial value to zero based on the training progress, as given by: $\eta = \eta_0 \cdot 0.5(\cos{(a \cdot \pi/2)} + 1)$. The newly added bottleneck and classifier layers have a learning rate 10 times that of the backbone. We set the initial learning rate for the backbone to 2e-4 and use a batch size of 128.

## E    THE USE OF LARGE LANGUAGE MODELS (LLMS)

Large language models (LLMs) were only used to improve the clarity, grammar, and fluency of the manuscript. They were not involved in the development of research ideas, experimental design, data analysis, or any other aspect of the scientific content.

