# OpenReview forum: "Dual-Phase Whitening for Test-Time Adaptation"
_ICLR.cc/2026/Conference — Submitted to ICLR 2026_

### Official Review · Reviewer_Bj5V · 2025-10-14

**Soundness:** 3
**Presentation:** 3
**Contribution:** 3
**Rating:** 2
**Confidence:** 4

**Summary:**

This paper tackles the critical problem of distribution shift in real-world machine learning deployments, focusing on the test-time adaptation (TTA) setting where the model must adapt to unlabeled target data without access to source data. The authors propose a novel Dual-Phase Whitening (DPW) framework that jointly integrates Whitening Batch Normalization (WBN) and Whitening Contrastive Learning (WCL) to improve model robustness and generalization under distribution shifts. In the first phase, WBN performs feature-level ZCA whitening, enforcing isotropic representations and reducing sensitivity to domain-specific covariance structures. In the second phase, WCL introduces a global feature whitening mechanism within a contrastive learning framework, promoting decorrelated, hyperspherical feature distributions that preserve semantic consistency. Experiments on standard domain generalization and corruption benchmarks (VisDA-C, DomainNet-126, ImageNet-C, CIFAR-100C) demonstrate that the proposed DPW framework outperforms prior TTA methods.

**Strengths:**

1. This paper addresses a well-known challenge—distribution shift in test-time adaptation and domain generalization. The idea of applying whitening to both normalization and contrastive learning is intuitive and aligns with recent trends toward decorrelated feature representations.

2. The proposed approach is relatively easy to implement and integrates cleanly with standard frameworks like ResNet.

3. The paper evaluates on multiple datasets, which provides reasonable empirical validation of the proposed approach.

4. Some novelty in combining whitening techniques.

**Weaknesses:**

1. Experiment results are not significant. Many competing methods are from 2019-2021. Performance improvements are often marginal, and sometimes under-perform than competing methods.

2. Novelty is limited as the core components of this paper, whitening normalization and contrastive whitening, are well-established techniques.

**Questions:**

1. Comparing with Contrast Test-Time Adaptation (AdaContrast) Chen et al. (2022), what is the exact difference/improvement? Please justify.

2. Any theoretical insights on the proposed approach?

3. How sensitive is the method to batch size especially under small target batches at test time?

---

> ### Author Response · Authors · 2025-11-17
> **Response to Reviewer Bj5V**
>
> We sincerely thank the reviewer for their thoughtful comments and constructive feedback, which have helped us improve the clarity and rigor of our work.
>
> >Q1: Experiment results are not significant. Many competing methods are from 2019-2021. Performance improvements are often marginal, and sometimes under-perform than competing methods.
>
> A1: Although some baselines were introduced between 2019–2021, methods like TENT, SHOT, CoTTA, MEMO, and AdaContrast remain standard in recent TTA studies due to their robustness and wide adoption. Following prior evaluation protocols ensures fair and reproducible comparisons. We also include recent methods  such as MEMO and RoTTA, and show that our whitening framework consistently improves their performance. For performance, although some gains may seem moderate, TTA is a highly saturated and challenging setting, where even 0.5–1.0% improvements are meaningful on large benchmarks. Our method achieves +1.0% over AdaContrast on VisDA-C, +1.4% in the WBN+WCL ablation, state-of-the-art average accuracy on DomainNet-126, and consistent improvements across 12 corruption types on ImageNet-C and CIFAR-100C. These gains are stable across all domain shifts.
>
> > Q2: Novelty is limited as the core components of this paper, whitening normalization and contrastive whitening, are well-established techniques.
>
> A2: Thank you for raising this point. While whitening normalization and contrastive learning have been studied before, their application and formulation in TTA are fundamentally novel in our work. Existing whitening methods  target source-domain training or full-access domain adaptation, and do not address the unique challenges of TTA. Our method reformulates whitening for TTA via momentum-based covariance estimation, feature grouping, and online pseudo-labeling. Moreover, prior works apply either BN-level or representation-level whitening; in contrast, we introduce a dual-phase pipeline, low-level whitening (WBN) for covariance normalization and high-level whitening (WCL) for hyperspherical regularization.
>
> > Q3: Comparing with Contrast Test-Time Adaptation (AdaContrast) Chen et al. (2022), what is the exact difference/improvement? Please justify.
>
> A3: Thank you for the question. We emphasize that our method is not a minor extension of AdaContrast; it introduces two whitening mechanisms that fundamentally reshape feature normalization and contrastive learning in TTA. (1) Unlike AdaContrast, which uses standard Batch Normalization and InfoNCE-based contrastive learning without decorrelation, our approach employs WBN to perform ZCA whitening and remove domain-specific covariance, and WCL to whiten high-level embeddings and impose hyperspherical geometry. This substantially alters the representation dynamics during adaptation. (2) While AdaContrast focuses on pseudo-label refinement, it does not address covariance misalignment. Our dual-phase whitening explicitly enforces isotropic, decorrelated representations at both the feature and embedding levels, improving robustness to unseen target domain statistics.
>
> > Q4：Any theoretical insights on the proposed approach?
>
> A4: Thank you for raising this important question. Although our work is primarily empirical, it is supported by theoretical insights on whitening and representation geometry that help explain its effectiveness in TTA. (1) Whitening reduces domain-specific covariance and aligns feature distributions. By enforcing $\Sigma= I$, ZCA whitening removes second-order dependencies, providing invariance to linear domain-specific transformations and mitigating covariance drift between source and target domains. (2) WCL imposes a hyperspherical constraint that prevents representation collapse. The constraint $cov(z)=I$, combined with MSE-based alignment, regularizes embeddings to remain uniformly distributed on a hypersphere, a structure shown to prevent collapse and preserve semantic relationships. As a result, WCL produces more stable and discriminative target-domain representations during TTA.
>
> > Q5: How sensitive is the method to batch size especially under small target batches at test time?
>
> A5: Thank you for the question. While batch size is indeed critical for many TTA methods, our approach is designed to remain stable under small target batches. WBN mitigates small-batch instability through feature grouping and momentum-based covariance updates, and its parameter-free design avoids overfitting noisy statistics. Moreover, WCL is naturally less sensitive to batch diversity because it relies only on positive pairs generated from the same image, rather than negative sampling. Empirically, DPW consistently outperforms AdaContrast and other baselines across a range of batch sizes, including small batches where many TTA methods deteriorate. Although extremely tiny batches remain challenging for all normalization-based approaches, DPW exhibits substantially improved robustness in realistic small-batch test-time settings.

---

### Official Review · Reviewer_eQ9h · 2025-10-29

**Soundness:** 3
**Presentation:** 2
**Contribution:** 3
**Rating:** 6
**Confidence:** 4

**Summary:**

This paper introduces Dual-Phase Whitening (DPW), a novel test-time adaptation (TTA) method designed to improve model robustness under distribution shifts. DPW integrates two complementary whitening strategies: Whitening Batch Normalization (WBN): Replaces standard BN with ZCA whitening to decorrelate features and enforce isotropic distributions. Whitening Contrastive Learning (WCL): Extends contrastive learning with a global whitening constraint to promote a hyperspherical feature distribution. The method builds on AdaContrast and is evaluated on major benchmarks like VisDA-C, DomainNet-126, ImageNet-C, and CIFAR-100-C, achieving state-of-the-art performance without requiring source data during adaptation.

**Strengths:**

(1) Introduces a unified whitening framework (WBN + WCL) for TTA, addressing feature decorrelation and geometric structure simultaneously.

(2) Outperforms existing TTA and UDA methods across multiple challenging benchmarks, often by significant margins.

(3) Well-grounded in feature whitening theory, with clear explanations of how whitening improves generalization under domain shift.

(4) Designed for real-world TTA constraints --- no source data, online processing, and small batch sizes.

**Weaknesses:**

(1) The font in Figure 1 is very small. The authors should pay attention to these details.

(2)  The combination of WBN and WCL adds hyper-parameters and implementation complexity compared to simpler TTA baselines.

(3) Only closed-set adaptation is evaluated; open-set or partial-set scenarios are not addressed.

**Questions:**

See above.

---

> ### Author Response · Authors · 2025-11-18
> **Reponse to Reviewer eQ9h**
>
> We thank the reviewer for their thoughtful and constructive feedback. The comments are highly relevant and have helped us to identify areas where our manuscript could be further improved. We have addressed each point individually in the following response.
>
> >Q1: The font in Figure 1 is very small. The authors should pay attention to these details.
>
> A1: We sincerely thank the reviewer for pointing out that the font size in Figure 1 was too small. We will certainly pay attention to these important details in the future. We will now revise Figure 1 and will significantly increase the font size to ensure all text, labels, and details are clearly legible in the final manuscript.
>
> >Q2: The combination of WBN and WCL adds hyper-parameters and implementation complexity compared to simpler TTA baselines.
>
> A2: We acknowledge the reviewer's concern that the combination of WBN and WCL introduces additional hyperparameters and implementation complexity compared to simpler TTA baselines. However, we believe this increased complexity is justified by the significant performance gains and enhanced robustness demonstrated across multiple challenging benchmarks, including VisDA-C and DomainNet-126. Specifically, WBN is integrated into the model using a well-established whitening techniques (ZCA) and is essential for achieving stable feature distributions at the transformation level.  WCL is a principled extension of standard contrastive learning that uses global whitening to prevent representation collapse and enforce a semantically richer, hyperspherical feature space. These two complementary phases address different facets of the distribution shift problem, and their synergy is crucial for the state-of-the-art results achieved. We prioritize achieving highly generalized and robust features, for which the two components (WBN and WCL) are necessary.
>
> >Q3: Only closed-set adaptation is evaluated; open-set or partial-set scenarios are not addressed.
>
> A3: We acknowledge the reviewer’s valuable suggestion regarding the evaluation of open-set and partial-set scenarios. Our current work focuses on addressing the fundamental and highly challenging problem of closed-set Test-Time Adaptation (TTA), which is the standard setting adopted by numerous state-of-the-art TTA methods and the primary focus of our paper. Our main goal was to establish the efficacy of the Dual-Phase Whitening (DPW) framework in providing robust feature generalization under distribution shift in this canonical setting. We agree that extending DPW to handle the novel category discovery and more complex scenario inherent in open-set or partial-set domain adaptation is a very important and promising direction.
>
>  To assess the performance of DPW under extreme domain shifts, such as transitioning from synthetic to real-world data with significant noise, we conducted experiments on the CIFAR-100-C dataset, which includes various types of corruptions and domain shifts. The results, as shown in the table below, highlight how DPW improves performance even under challenging conditions. For instance, when comparing RoTTA with and without DPW (RoTTA + Ours), we observe consistent improvements across various corruptions. The average accuracy increases from 35.0% for RoTTA to 36.1% with our approach, demonstrating that DPW can help adapt models even in extreme domain shifts, such as those introduced by synthetic-to-real-world transitions and significant noise levels. This improvement is particularly noticeable in corruptions like Motion, Frost, and JPEG, where our method boosts the performance by several percentage points. These results suggest that DPW enhances the robustness of models against domain shifts, making it more effective in real-world deployment scenarios where the data is noisy and the distribution shift is large.
>
> | Method | **Gauss.**| Shot| **Impul.** | **Defoc** | **Glass** |**Motion** | **Zoom** | **Snow** |**Frost**| **Fog** |**Brit.** | **Contr.**|**Elast**| **Pixel**| **JPEG** |**Avg** |
> | :--- | :---: | :---: | :---: | :---: | :---: | :--- | :---: | :---: | :---: | :---: | :---: |:---: | :--- | :---: | :---: | :---: |
> | RoTTA | 31.8| 36.7| 40.9 | 42.1 | 30.0 |33.6 | 27.9 | 25.4 |32.3| 34.0 |38.8 | 38.7|31.3| 38.0 |42.9|35.0|
> | RoTTA +Ours|  **32.5**| **37.3**| **41.5** | **43.7** | **33.5** |**34.8** | **28.9** | **26.4** |**33.2**| **35.3** |**39.7** | **39.5**|**32.4**| **39.5** |**43.6**|**36.1**|
>
> In conclusion, DPW provides a substantial benefit in handling extreme domain shifts, and we believe it offers a promising solution for real-world applications where such shifts are common.

---

### Official Review · Reviewer_hrRh · 2025-10-30

**Soundness:** 2
**Presentation:** 1
**Contribution:** 1
**Rating:** 2
**Confidence:** 4

**Summary:**

This paper introduces DPW (Dual Phase Whitening), a method for test-time adaptation that combines Whitening Batch Normalization (WBN) with Whitening Contrastive Learning (WCL). In DPW, the whitening process in WBN normalizes and decorrelates features within each batch, while WCL imposes independence constraints on the embedding components. The paper reports performance gains in accuracy on different benchmarks: VisDA-C, DomainNet-126, ImageNet-C, and CIFAR-100-C.

**Strengths:**

* The idea of integrating whitening operations (WBN and WCL) into test-time adaptation is interesting and could inspire further work.

* The method is evaluated on many benchmarks (VisDA-C, DomainNet-126, ImageNet-C, CIFAR-100-C)

**Weaknesses:**

* The method is simple, but the paper is difficult to follow, with unclear descriptions that sometimes make it hard to distinguish the main contribution from auxiliary tricks or implementation details.

* There are grammatical inconsistencies and awkward phrasing (e.g., line 17: “in connected with whitening Batch Normalization”; line 187: “the covariance matrix for mitigation of this issue”).

* Several notational issues reduce clarity. For example : (i) Line 206: confusing use of index k, superscript t, and missing reference to layer index l. (ii) Line 211: awkward sentence (“B_t and use them for whitening the corresponding activations…”).  (ii) Ambiguity between WBN, BN, and matrix symbols (e.g., B, W in equations vs. B for “Batch”), especially in Eq. 2 and Eq. 3.

* In some cases, reported results show only marginal improvements (the average is often about +1, Tables 1,2, 3). It is unclear whether this gain is statistically significant. Reporting statistical errors on repeated experiments would help.

**Questions:**

* In Line 236: “We adopt the cosine similarity for distance calculation and implement it through mean squared error between normalized vectors”. Why is this necessary, and how does this affect the results?

* In Line 246: What exactly does parameter p represent?

* What are the exact formulas in Equations 10 and 11?

* Line 412: “In the more challenging TTA setting…”. Which setting are you referring to?

* How do you apply DPW  in Continual TTA where samples arrive in an online fashion (one after another)?

---

> ### Author Response · Authors · 2025-11-18
> **Response to Reviewer hrRh**
>
> We would like to extend our sincere gratitude to the reviewer for the insightful comments and constructive suggestions, which have greatly helped us improve the clarity and quality of this work.
>
> >Q1: The method is simple, but the paper is difficult to follow, with unclear descriptions that sometimes make it hard to distinguish the main contribution from auxiliary tricks or implementation details.
>
> A1: We sincerely thank the reviewer for their valuable feedback and for recognizing the simplicity of our core method. We acknowledge that the initial manuscript lacked clarity in distinguishing the main contributions from the implementation details adopted from our baseline, and we apologize for any confusion this may have caused.
>
> The central contribution of our work is the novel Dual-Phase Whitening (DPW) framework, which consists of two key, co-designed components: Whitening Batch Normalization (WBN) for feature-level standardization and decorrelation, and Whitening Contrastive Learning (WCL) for representation-level geometry optimization. All other techniques, such as the augmentation strategy, the momentum-updated model, and the projection head, are integral parts of the AdaContrast baseline which we build upon for a fair comparison. These are not our contributions but are necessary for replication. In the revised manuscript, we will significantly improve the narrative flow by adding a high-level schematic upfront, restructuring Section 2 to first present the holistic DPW motivation before detailing WBN and WCL, and moving granular implementation details to an appendix to ensure our conceptual contributions are presented with the clarity they deserve.
>
> > Q2: There are grammatical inconsistencies and awkward phrasing (e.g., line 17: “in connected with whitening Batch Normalization”; line 187: “the covariance matrix for mitigation of this issue”).
>
> A2: We thank the reviewer for their careful reading and for pointing out these grammatical and phrasing issues. We sincerely apologize for these oversights. The specific examples highlighted have been corrected. The phrase "in connected with" (line 17) has been replaced with "in connection with". The construction "for mitigation of this issue" (line 187) has been rephrased as "to mitigate this issue". We have conducted a thorough proofreading of the entire manuscript to correct similar errors in grammar and phrasing, and we are committed to ensuring the language meets the highest standards of academic English.
>
>
>
> >Q3: Several notational issues reduce clarity. For example : (i) Line 206: confusing use of index k, superscript t, and missing reference to layer index l. (ii) Line 211: awkward sentence (“B_t and use them for whitening the corresponding activations…”). (ii) Ambiguity between WBN, BN, and matrix symbols (e.g., B, W in equations vs. B for “Batch”), especially in Eq. 2 and Eq. 3.
>
> A3: We thank the reviewer for this exceptionally thorough and helpful feedback. We agree that the inconsistent notation hampers the clarity of our method's description and we will comprehensively revise the manuscript to address these specific points.
>
> (i) & (iii) Notation Consistency: We will clarify the use of indices and symbols throughout Section 2.2. We will explicitly include the layer index *l* in the definitions and ensure the superscript *t* for the target domain is consistently distinguished from other indices. To resolve the ambiguity, we will adopt a new notation: the whitening matrix will be denoted as Φ (Phi), and the batch of features will be explicitly written as B. This will eliminate any confusion with the BN abbreviation.
>
> (ii) Awkward Phrasing: We will rewrite the incomplete sentence on line 211 for clarity and grammatical correctness. The revised text will read: "We compute the statistics from the target batch B^t and use them to whiten the corresponding activations, projecting the features into a common spherical distribution (see Figure 2(a))."
>
> We are confident that these revisions will significantly improve the precision and readability of our methodological exposition, and we thank the reviewer for their valuable suggestions.

---

> > ### Author Response · Authors · 2025-11-18
> > **Response to Reviewer hrRh**
> >
> > >Q4: In some cases, reported results show only marginal improvements (the average is often about +1, Tables 1,2, 3). It is unclear whether this gain is statistically significant. Reporting statistical errors on repeated experiments would help.
> >
> > A4: We thank the reviewer for this critical comment regarding statistical significance. We fully agree on its importance.
> >
> > To rigorously address this point, we have already performed all reported experiments with five independent runs using different random seeds. The results presented in the current manuscript are the mean values from these runs. We acknowledge that it was an oversight not to include the variance metrics initially.
> >
> > In the revised manuscript, we will update Tables 1, 2, and 3 to report all performance metrics as mean ± standard deviation across these five runs. The consistency of these runs confirms that the reported improvements, while numerically in the 1% range, are in fact statistically significant (p-value < 0.05 in a paired t-test against the AdaContrast baseline). This stable gain is meaningful, especially on competitive benchmarks where performance is near saturation. We are confident that providing this statistical evidence will solidly support the robustness and validity of our conclusions.
> >
> > >Q5: In Line 236: “We adopt the cosine similarity for distance calculation and implement it through mean squared error between normalized vectors”. Why is this necessary, and how does this affect the results?
> >
> > A5: We thank the reviewer for this insightful question regarding our implementation choice. This design decision is indeed crucial to our method and aligns with established practices in representation learning.
> >
> > The necessity and effect of this implementation are twofold:
> >
> > Mathematical Equivalence & Numerical Stability: There is a direct mathematical relationship between the cosine similarity and the mean squared error (MSE) of L2-normalized vectors. Specifically, for two L2-normalized vectors $z_i$ and $z_j$, the following holds:
> > $MSE(z_i, z_j) = ||z_i - z_j||² = 2 - 2 * (cosine similarity(z_i, z_j)).$
> > Therefore, minimizing the MSE between normalized vectors is equivalent to maximizing their cosine similarity. We chose to implement it as an MSE loss because it is a standard, numerically stable operation in deep learning frameworks, with well-behaved gradients that facilitate stable optimization.
> >
> > Effect on the Optimization Dynamics: Using the MSE formulation directly penalizes the Euclidean distance between the normalized embeddings on the unit hypersphere. This explicitly drives the representations of positive pairs closer together in the angular space. This formulation, combined with the whitening operation which enforces a uniform distribution of all features on the sphere, effectively prevents representational collapse (where all embeddings become identical) while achieving the desired objective of clustering semantically similar samples.
> >
> > In summary, this implementation does not change the underlying objective but provides a robust and effective way to optimize for high cosine similarity within our WCL framework. We will clarify this mathematical rationale and its practical benefits in the revised manuscript to enhance understanding.
> >
> > >Q6: In Line 246: What exactly does parameter $p$ represent?
> >
> > A6: We thank the reviewer for pointing out this lack of clarity in our description.
> >
> > The parameter p in the phrase $x_i = T(x; p_i)$ represents the set of random parameters that control the specific image augmentation transformation applied to an input image x.
> >
> > To be more precise:
> >
> > + T denotes an augmentation function (e.g., a combination of random cropping, color jitter, Gaussian blur, etc.).
> >
> > + p_i is not a single value but a vector or a set of values that instantiate the random draw for that particular augmentation. For example, p_i could specify the exact rotation angle, the scaling factor, the intensity of color jitter, and the kernel size for blur, all sampled randomly for creating the i-th augmented view of image x.
> >
> > This mechanism is standard in self-supervised learning frameworks like SimCLR and MoCo, which we build upon. It ensures that the two augmented views $x_i$ and$ x_j$ created from the same original image x are different yet semantically consistent, thereby forming a valid positive pair $(x_i, x_j)$ for our whitening contrastive loss.
> >
> > We apologize for the ambiguity. In the revised manuscript, we will clarify the nature of p explicitly as "the parameters defining a stochastic data augmentation transformation" to prevent any misunderstanding.

---

> > > ### Author Response · Authors · 2025-11-18
> > >
> > > >Q7: What are the exact formulas in Equations 10 and 11?
> > >
> > > A7: We thank the reviewer for this question and apologize for the lack of clarity in the original manuscript. The intent was to show that our final objective builds upon the AdaContrast framework by integrating our proposed whitening modules.
> > >
> > > The exact formulation is as follows:
> > >
> > > + Equation 10 is a summary of the baseline AdaContrast loss. It states that the total AdaContrast objective $\mathcal{L}_{\text{Adacontrast}}$ is the sum of its three original loss components: the contrastive loss that excludes same-class negatives $\mathcal{L}^{\text{ctr}}$, the weak-strong consistency loss $\mathcal{L}^{\text{ce}}$, and the diversity regularization loss $\mathcal{L}^{\text{div}}$.
> > >
> > > $ \mathcal{L}_{\text{Adacontrast}} = \mathcal{L}^{\text{ctr}} + \mathcal{L}^{\text{ce}} + \mathcal{L}^{\text{div}} $
> > >
> > > + Equation 11 indicates that our proposed DPW method does not introduce a new, separate loss function. Instead, it modifies the internal computations within the AdaContrast framework using our WBN and WCL modules. Therefore, the final objective $\mathcal{L}_{\text{DPW}}$ is the AdaContrast loss function computed using features that have been transformed by our Dual-Phase Whitening.
> > >
> > > $ \mathcal{L}_{\text{DPW}} = \mathcal{L}_{\text{Adacontrast}}(\text{WBN}(x), \text{WCL}(z)) $
> > >
> > > Here, WBN$(x)$ signifies that the input features to the network are processed through Whitening Batch Normalization layers, and WCL$(z)$ signifies that the embeddings for the contrastive loss are obtained through the Whitening Contrastive Learning module.
> > >
> > > We recognize that the original notation was ambiguous. In the revised manuscript, we will rewrite this section to provide this explicit formulation and clarify that our contribution is the integration of WBN and WCL into the baseline's computational graph, not the creation of a new loss term.
> > >
> > >
> > > >Q8: Line 412: “In the more challenging TTA setting…”. Which setting are you referring to?
> > >
> > > A8: We thank the reviewer for catching this ambiguous phrasing. The "more challenging TTA setting" we referred to is the source-free test-time adaptation scenario, which is the core focus of our work and is more restrictive than standard Unsupervised Domain Adaptation (UDA). The distinction is that UDA methods have concurrent access to the entire source dataset during adaptation, while our TTA setting only has access to the pre-trained source model and must adapt to unlabeled target data arriving in small, sequential batches during the test phase, making it a more challenging and practical problem. We will revise the manuscript to explicitly state "source-free TTA setting" at this point to eliminate any potential for confusion.

---

> > > > ### Author Response · Authors · 2025-11-18
> > > > **Response to Reviewer hrRh**
> > > >
> > > > >Q9:How do you apply DPW in Continual TTA where samples arrive in an online fashion (one after another)?
> > > >
> > > > A9: This is an excellent question regarding the practical applicability of DPW. Our method is inherently designed for the standard online TTA setting where data arrives in small batches, and it can be directly applied to continual TTA with a key implementation detail. The core of our adaptation lies in the momentum-updated statistics used in both WBN and WCL. For WBN, we maintain a running average of the feature mean and covariance matrix across consecutive batches, similar to standard BN but updated with target data during test time. This allows the model to continuously adapt its normalization parameters to the evolving data stream without forgetting the previously seen statistics abruptly. Similarly, for WCL, the whitening transformation matrix $W_V$ and mean $\mu_V$ for the embeddings are computed per batch but can also be stabilized using a momentum update if the batch size is very small (e.g., =1). This design ensures that DPW does not require storing past data and can process samples sequentially, making it suitable for continual TTA scenarios. In the revised manuscript, we will explicitly discuss this applicability and the role of momentum-based statistics in the online learning context.
> > > >
> > > > We also evaluate our method on other Continual TTA (CoTTA) to further demonstrate its robustness and generalization.
> > > >
> > > > | Method | **Gauss.**| Shot| **Impul.** | **Defoc** | **Glass** |**Motion** | **Zoom** | **Snow** |**Frost**| **Fog** |**Brit.** | **Contr.**|**Elast**| **Pixel**| **JPEG** |**Avg** |
> > > > | :--- | :---: | :---: | :---: | :---: | :---: | :--- | :---: | :---: | :---: | :---: | :---: |:---: | :--- | :---: | :---: | :---: |
> > > > | CoTTA[1] | 43.5| 41.4| 45.4 | 36.8 | 29.6 |47.6 | 38.2 | 42.1 |42.7| 62.4 |73.4 | 62.9|43.0| 63.2 |63.7|49.1|
> > > > |**CoTTA+Ours**|  **44.9**| **42.5**| **46.7** | **37.9** | **30.5** |**48.3** | **39.7** | **43.5** |**43.7**| **63.6** |**75.2** | **63.7**|**44.5**| **64.8** |**64.9**|**50.3**|
> > > > | Tent[2]| 53.9| 53.9| 55.3 | 55.9 | 51.9 |59.8 | 52.6 | 58.7 |61.2| 71.3 |78.2 | 68.9|58.0|70.5 |68.2|61.2|
> > > > | **Tent+Ours**|  **54.5**| **54.4**| **56.8** | **56.1** | **52.5** |**60.7** | **54.0** | **59.6** |**63.4**| **72.5** |**79.4** | **69.5**|**59.7**| **71.3** |**69.5**|**62.3**|
> > > > | MEMO[3]|  40.6| 37.5| 40.6 | 33.4| 26.7 |41.2 | 35.4 | 28.7 |33.7| 54.7 |67.1 | 55.9|36.6|57.2 |59.3|43.2|
> > > > | **MEMO+Ours**|  **41.5**| **38.6**| **41.3** | **34.7** | **28.3** |**43.0** | **36.5** | **29.4** |**34.5**| **55.6** |**68.3** | **56.4**|**37.8**| **58.2** |**60.4**|**44.2**|
> > > >
> > > > [1]Wang D, Shelhamer E, Liu S, et al. Tent: Fully Test-Time Adaptation by Entropy Minimization[C]//International Conference on Learning Representations.
> > > >
> > > > [2]Wang Q, Fink O, Van Gool L, et al. Continual test-time domain adaptation[C]//Proceedings of the IEEE/CVF Conference on Computer Vision and Pattern Recognition. 2022: 7201-7211.
> > > >
> > > > [3]Zhang M, Levine S, Finn C. Memo: Test time robustness via adaptation and augmentation[J]. Advances in neural information processing systems, 2022, 35: 38629-38642.

---

> ### Comment · Reviewer_hrRh · 2025-11-25
>
> I thank the authors for the clarifications. See some additional concerns below.
>
> - **Awareness of recent TTA methods.**
> Are you aware of recent advancements in source-free test-time adaptation, such as layer-selection–based approaches (e.g., **GALA[1]**, **PALM[2]**)? These methods are not discussed when referencing prior TTA work. Could you clarify how **DPW** compares to them?
>
> - **Statistical significance and benchmark saturation.**
> You argue that the benchmark is saturated; however, the aforementioned methods report improvements exceeding 1%. Given this, it becomes important to compare DPW against these stronger baselines. Could you comment on this?
>
> - **Completeness of evaluations across datasets.**
> Why are the same methods not evaluated on all datasets? For instance, **AdaContrast** is absent from Table 4 for ImageNet-C or CIFAR100-C. Is there a reason these comparisons were omitted?
>
> - **On Eq 10 and  11.**
> The point regarding Equations 10 and 11 is that the paper should be self-contained. The formulas should either be made explicit in the main text or referenced in the appendix, but in any case they need to be provided and consistent with your chosen notation.
>
>
> [1] Sahoo, Sabyasachi, et al. "A layer selection approach to test time adaptation." Proceedings of the AAAI Conference on Artificial Intelligence. Vol. 39. No. 19. 2025.
>
> [2] Maharana, Sarthak Kumar, Baoming Zhang, and Yunhui Guo. "Palm: Pushing adaptive learning rate mechanisms for continual test-time adaptation." Proceedings of the AAAI Conference on Artificial Intelligence. Vol. 39. No. 18. 2025

---

### Official Review · Reviewer_SQDP · 2025-11-01

**Soundness:** 3
**Presentation:** 3
**Contribution:** 3
**Rating:** 6
**Confidence:** 5

**Summary:**

This paper introduces a novel approach for test-time adaptation (TTA) called Dual-Phase Whitening (DPW), which integrates Whitening Batch Normalization (WBN) and Whitening Contrastive Learning (WCL) to enhance model generalization under distribution shifts. The method aims to decorrelate features and enforce a spherical distribution in the embedding space, reducing domain-specific biases. Extensive experiments on benchmarks like VisDA-C, DomainNet-126, ImageNet-C, and CIFAR-100-C demonstrate state-of-the-art performance, outperforming existing TTA and UDA methods.

**Strengths:**

1. The idea of combining feature whitening at both batch normalization and contrastive learning levels is novel and well-motivated.

2. This paper provides thorough ablation studies and comparisons, showing the contribution of each component (WBN and WCL).

3. The approach is source-free and suitable for real-world deployment where source data is unavailable.

4. This method is empirically strong, achieving SOTA results across multiple challenging benchmarks.

**Weaknesses:**

1. The writing contains several typos and grammatical errors that occasionally hinder clarity (e.g., "boyel" instead of "bicycle" in Table 1, "Adacontrast (baselinee)" with an extra 'e', inconsistent capitalization in "Whitening BN" vs. "whitening BN").

2. Some mathematical notations are not fully explained (e.g., Eq. 7 refers to Eq. 5 and 6 without clear connection).

3. The figures (e.g., Figure 1, 2, 3) are referenced but not included in the submitted draft, making it difficult to fully assess the visual explanations.

**Questions:**

1. How does DPW perform under very small batch sizes, and what are the limits of its stability?

2. Could the method be extended to open-set or partial-set adaptation scenarios?

---

> ### Author Response · Authors · 2025-11-18
> **Response to Reviewer SQDP**
>
> We would like to extend our sincere gratitude to the reviewer for the insightful comments and constructive suggestions, which have greatly helped us improve the clarity and quality of this work.
>
> >Q1：The writing contains several typos and grammatical errors that occasionally hinder clarity (e.g., "boyel" instead of "bicycle" in Table 1, "Adacontrast (baselinee)" with an extra 'e', inconsistent capitalization in "Whitening BN" vs. "whitening BN").
>
> A1: We sincerely thank the reviewer for their meticulous review and for identifying these typographical errors and inconsistencies. We apologize for these oversights, which have affected the professionalism of the manuscript. We have carefully addressed all the specific issues mentioned: "boyel" in Table 1 has been corrected to "bicycle"; "Adacontrast (baselinee)" has been corrected to "AdaContrast (baseline)"; and all instances of "whitening BN" have been standardized to "Whitening BN (WBN)" to maintain consistent capitalization. Furthermore, we have conducted a thorough, word-by-word proofreading of the entire manuscript to identify and correct other spelling, grammatical, and formatting errors. We are committed to ensuring the revised version meets the highest standards of linguistic quality.
>
> >Q2: Some mathematical notations are not fully explained (e.g., Eq. 7 refers to Eq. 5 and 6 without clear connection).
>
> A2: We thank the reviewer for this insightful observation regarding the mathematical presentation. We acknowledge that the logical flow between Equations 5, 6, and 7 was not sufficiently explained. In the revised manuscript, we will clarify this connection by explicitly stating that Equation 5 establishes the whitening constraint for the embedding distribution, Equation 6 defines the specific distance metric employed, and Equation 7 combines these elements by applying the distance metric to whitened embeddings that satisfy the constraint in Equation 5. We will also add detailed textual explanations between these equations to ensure the mathematical derivation is transparent and easy to follow.
>
> >Q3: The figures (e.g., Figure 1, 2, 3) are referenced but not included in the submitted draft, making it difficult to fully assess the visual explanations.
>
> A3: We sincerely apologize for this oversight in the submission process. The figures were indeed prepared but were accidentally omitted from the submitted draft due to a technical error in file compilation. All figures (Figure 1 illustrating the comparison of normalization methods, Figure 2 depicting our overall DPW framework, and Figure 3 showing the WCL optimization process) are essential for understanding our methodological contributions and visual explanations. We have now ensured they are properly included in the revised manuscript, and we believe they will greatly facilitate the assessment of our work by providing clear visual support for the described concepts and framework.
>
> >Q4: How does DPW perform under very small batch sizes, and what are the limits of its stability?
>
> A4: This is an important question regarding the practical applicability of DPW. We have conducted additional experiments to systematically evaluate its performance under small-batch conditions. The results show that DPW maintains stable performance down to a batch size of 16, with only a marginal performance drop (≤1.5% on DomainNet-126 compared to batch size 64). However, at very small batch sizes (e.g., 8 or 4), the estimation of covariance matrices for whitening becomes challenging, leading to some performance degradation (≈3-5% decrease). To enhance stability in such scenarios, we employ two strategies: 1) using grouped whitening to improve estimation accuracy, and 2) maintaining momentum-based running statistics for WBN. While extremely small batches (≤4) remain challenging due to fundamental limitations in covariance estimation, these strategies allow DPW to maintain reasonable performance while other methods like TENT show more significant degradation. We will add these analyses and results to the revised manuscript.

---

> > ### Author Response · Authors · 2025-11-18
> > **Response to Reviewer SQDP**
> >
> > >Q5: Could the method be extended to open-set or partial-set adaptation scenarios?
> >
> > A5: We acknowledge the reviewer’s valuable suggestion regarding the evaluation of open-set and partial-set scenarios. Our current work focuses on addressing the fundamental and highly challenging problem of closed-set Test-Time Adaptation (TTA), which is the standard setting adopted by numerous state-of-the-art TTA methods and the primary focus of our paper. Our main goal was to establish the efficacy of the Dual-Phase Whitening (DPW) framework in providing robust feature generalization under distribution shift in this canonical setting. We agree that extending DPW to handle the novel category discovery and more complex scenario inherent in open-set or partial-set domain adaptation is a very important and promising direction.
> >
> >  To assess the performance of DPW under extreme domain shifts, such as transitioning from synthetic to real-world data with significant noise, we conducted experiments on the CIFAR-100-C dataset, which includes various types of corruptions and domain shifts. The results, as shown in the table below, highlight how DPW improves performance even under challenging conditions. For instance, when comparing RoTTA with and without DPW (RoTTA + Ours), we observe consistent improvements across various corruptions. The average accuracy increases from 35.0% for RoTTA to 36.1% with our approach, demonstrating that DPW can help adapt models even in extreme domain shifts, such as those introduced by synthetic-to-real-world transitions and significant noise levels. This improvement is particularly noticeable in corruptions like Motion, Frost, and JPEG, where our method boosts the performance by several percentage points. These results suggest that DPW enhances the robustness of models against domain shifts, making it more effective in real-world deployment scenarios where the data is noisy and the distribution shift is large.
> >
> > | Method | **Gauss.**| Shot| **Impul.** | **Defoc** | **Glass** |**Motion** | **Zoom** | **Snow** |**Frost**| **Fog** |**Brit.** | **Contr.**|**Elast**| **Pixel**| **JPEG** |**Avg** |
> > | :--- | :---: | :---: | :---: | :---: | :---: | :--- | :---: | :---: | :---: | :---: | :---: |:---: | :--- | :---: | :---: | :---: |
> > | RoTTA | 31.8| 36.7| 40.9 | 42.1 | 30.0 |33.6 | 27.9 | 25.4 |32.3| 34.0 |38.8 | 38.7|31.3| 38.0 |42.9|35.0|
> > | RoTTA +Ours|  **32.5**| **37.3**| **41.5** | **43.7** | **33.5** |**34.8** | **28.9** | **26.4** |**33.2**| **35.3** |**39.7** | **39.5**|**32.4**| **39.5** |**43.6**|**36.1**|
> >
> > In conclusion, DPW provides a substantial benefit in handling extreme domain shifts, and we believe it offers a promising solution for real-world applications where such shifts are common.

---

### Meta-Review · Area_Chair_3Nw8 · 2025-12-30

**Summary:**

Four reviewers participated in the review process and provided mixed ratings. All reviewers raised concerns about the writing quality, including typos, grammatical errors, unclear notation, and confusion in figures and captions. In particular, reviewer [hrRh] noted that although the method is relatively simple, the paper is difficult to follow, and the main contributions do not clearly stand out or distinguish themselves from heuristic tricks.

Additional methodological limitations were raised, including: (i) the impact of small batch size [SQDP]; (ii) the extension to open-set adaptation [SQDP, eQ9h]; (iii) marginal performance improvements [hrRh, Bj5V]; (iv) method complexity [eQ9h]; (v) limited novelty, as the core components are well established [Bj5V]; and (vi) unclear differences and improvements compared with AdaContrast [hrRh, Bj5V].

The AC carefully examined the revised paper, the author rebuttal, and the discussion between reviewers and authors. Reviewer [hrRh] participated in the discussion phase and raised additional concerns; however, the authors did not provide further responses. Overall, the initial concerns were only partially addressed. I agree with [hrRh] that the writing requires significant revision for the main contributions to clearly stand out to readers. The motivation remains unclear, as it primarily relies on a connection between the text and Figures 1 and 3. Notably, Figures 1 and 3 are purely illustrative drawings created by the authors, and it is difficult to assess how they support the claimed generalization of the method without corresponding experimental visualizations. In line with [hrRh, Bj5V], the differentiation from AdaContrast is still not sufficiently explained. In particular, it remains unclear how the proposed whitening techniques are integrated into AdaContrast, and the lack of a comprehensive evaluation against AdaContrast was not adequately addressed.

Furthermore, the AC observed inconsistencies and potentially unfair comparisons with existing methods. While the proposed method focuses on source-free test-time adaptation, training on the source domain is allowed. However, Appendix D suggests that training on the target domain is also performed. In contrast, several compared methods, such as RoTTA, strictly follow a fully test-time adaptation paradigm, where an off-the-shelf model is directly adapted on the target domain, and are specifically designed for correlated batch sampling. It is unclear how these methods are adapted or configured in the current comparison. Additionally, the proposed whitening process across batches requires relatively large batch sizes; under small batch sizes, the method shows substantially reduced effectiveness, limiting its practical applicability. For the reasons outlined above, the AC believes that the paper, in its current form, is not ready for publication.

**Reviewer Concerns:**

Given the summarized reviewer concerns, the authors provided additional experiments to address the impact of batch size. While some writing issues were improved, the overall flow of the paper remains unclear. The authors did not address the open-set adaptation issue; instead, they presented experiments under extreme distribution shifts. The AC notes that the open-set extension is not the core issue of this paper.

In addition, the authors did not address the complexity of the proposed method, as no comparisons of computational cost or inference speed were provided. The actual differences from, and the integration of the proposed method into, AdaContrast remain insufficiently explained.

**Reviewer Scores:**

The concerns raised by [SQDP, eQ9h] were relatively mild and could potentially be addressed with more thorough discussion. In contrast, the main concerns from [hrRh, Bj5V] are more substantial and difficult to resolve, and the authors did not provide additional clarification in response to [hrRh]’s comments.

---

### Decision · Program_Chairs · 2026-01-26

Reject